# A FINE-GRAINED ANALYSIS ON DISTRIBUTION SHIFT

**Olivia Wiles    Sven Gowal    Florian Stimberg    Sylvestre-Alvise Rebuffi**
**Ira Ktena    Krishnamurthy (Dj) Dvijotham    Taylan Cemgil**
DeepMind, London, UK

`{oawiles,sgowal,stimberg,sylvestre,iraktena,taylancemgil}@deepmind.com    dvij@google.com`

## ABSTRACT

Robustness to distribution shifts is critical for deploying machine learning models in the real world. Despite this necessity, there has been little work in defining the underlying mechanisms that cause these shifts and evaluating the robustness of algorithms across multiple, different distribution shifts. To this end, we introduce a framework that enables fine-grained analysis of various distribution shifts. We provide a holistic analysis of current state-of-the-art methods by evaluating 19 distinct methods grouped into five categories across both synthetic and real-world datasets. Overall, we train more than 85K models. Our experimental framework can be easily extended to include new methods, shifts, and datasets. We find, unlike previous work (Gulrajani & Lopez-Paz, 2021), that progress has been made over a standard ERM baseline; in particular, pretraining and augmentations (learned or heuristic) offer large gains in many cases. However, the best methods are not consistent over different datasets and shifts. Code is available at github.com/deepmind/distribution_shift_framework.

## 1 INTRODUCTION

If machine learning models are to be ubiquitous in critical applications such as driverless cars (Janai et al., 2020), medical imaging (Erickson et al., 2017), and science (Jumper et al., 2021), it is pivotal to build models that are robust to distribution shifts. Otherwise, models may fail surprisingly in ways that derail trust in the system. For example, Koh et al. (2020); Perone et al. (2019); AlBadawy et al. (2018); Heaven (2020); Castro et al. (2020) find that a model trained on one set of hospitals may not generalise to the imaging conditions of another; Alcorn et al. (2019); Dai & Van Gool (2018) find that a model for driverless cars may not generalise to new lighting conditions or object poses; and Buolamwini & Gebru (2018) find that a model may perform worse on subsets of the distribution, such as different ethnicities, if the training set has an imbalanced distribution. Thus, it is important to understand when we expect a model to generalise and when we do not. This would allow a practitioner to have confidence in the system (e.g. if a model is demonstrated to be robust to the imaging conditions of different hospitals, then it can be deployed in new hospitals with confidence).

While domain generalization is a well studied area, Gulrajani & Lopez-Paz (2021); Schott et al. (2021) have cast doubt on the efficacy of existing methods, raising the question: *has any progress been made in domain generalization over a standard expectation risk minimization (ERM) algorithm?* Despite these discouraging results, there are many examples that machine learning models *do* generalise across datasets with different distributions. For example, CLIP (Radford et al., 2021), with well engineered prompts, generalizes to many standard image datasets. Taori et al. (2020) found that models trained on one image dataset generalise to another, albeit with some drop in performance; in particular, higher performing models generalise better. However, there is little understanding and experimentation on *when* and *why* models generalise, especially in realistic settings inspired by real-world applications. This begs the following question:

*Can we define the important distribution shifts to be robust to and then systematically evaluate the robustness of different methods?*

To answer the above question, we present a grounded understanding of robustness to distribution shifts. We draw inspiration from disentanglement literature (see section 6), which aims to separate images into an independent set of factors of variation (or attributes). In brief, we assume the data

is composed of some (possibly extremely large) set of attributes. We expect models, having seen some distribution of values for an attribute, to be able to learn invariance to that attribute and so to generalise to unseen examples of the attribute and different distributions over that attribute. Using a simple example to clarify the setup, assume our data has two attributes (shape and color) among others. Given data with some distribution over the set of possible colors (e.g. red and blue) and the task of predicting shape (e.g. circle or square), we want our model to generalise to unseen colors (e.g. green) or a different distribution of colors (e.g. there are very few red circles in the training set, but the samples at evaluation are uniformly sampled from the set of possible colors and shapes).

Using this framework, we evaluate models across three distribution shifts: *spurious correlation*, *low-data drift*, and *unseen data shift* (illustrated in figure 1) and two additional conditions (label noise and dataset size). We choose these settings as they arise in the real world and harm generalization performance. Moreover, in our framework, these distribution shifts are the fundamental blocks of building more complex distribution shifts. We additionally evaluate models when there is varying amounts of label noise (as inspired by noise arising from human raters) and when the total size of the train set varies (to understand how models perform as the number of training examples changes). The unique ability of our framework to evaluate fine-grained performance of models across different distribution shifts and under different conditions is of critical importance to analyze methods under a variety of real-world settings. This work makes the following contributions:

- We propose a framework to define when and why we expect methods to generalise. We use this framework to define three, real world inspired distribution shifts. We then use this framework to create a systematic evaluation setup across real and synthetic datasets for different distribution shifts. Our evaluation framework is easily extendable to new distribution shifts, datasets, or methods to be evaluated.

- We evaluate and compare 19 different methods (training more than 85K models) in these settings. These methods span the following 5 common approaches: architecture choice, data augmentation, domain generalization, adaptive algorithms, and representation learning. This allows for a direct comparison across different areas in machine learning.

- We find that simple techniques, such as data augmentation and pretraining are often effective and that domain generalization algorithms do work for certain datasets and distribution shifts. However, there is no easy way to select the best approach a-priori and results are inconsistent over different datasets and attributes, demonstrating there is still much work to be done to improve robustness in real-world settings.

## 2 FRAMEWORK TO EVALUATE GENERALIZATION

In this section we introduce our robustness framework for characterizing distribution shifts in a principled manner. We then relate three common, real world inspired distribution shifts.

### 2.1 LATENT FACTORISATION

We assume a joint distribution $p$ of inputs $\boldsymbol{x}$ and corresponding attributes $y^1, y^2, \ldots, y^K$ (denoted as $y^{1:K}$) with $y^k \in \mathbb{A}^k$ where $\mathbb{A}^k$ is a finite set. One of these $K$ attributes is a label of interest, denoted as $y^l$ (in a mammogram, the label could be cancer/benign and a nuisance attribute $y^i$ with $i \neq l$ could be the identity of the hospital where the mammogram was taken). Our aim is to build a classifier $f$ that minimizes the risk $R$. However, in real-world applications, we only have access to a finite set of inputs and attributes of size $n$. Hence, we minimize the empirical risk $\hat{R}$ instead:

$$R(f) = \mathbb{E}_{(\boldsymbol{x}, y^l) \sim p} \left[ \mathcal{L}(y^l, f(\boldsymbol{x})) \right] \qquad \hat{R}(f; p) = \frac{1}{n} \sum_{\{(y_i^l, \boldsymbol{x}_i) \sim p\}_{i=1}^n} \mathcal{L}(y_i^l, f(\boldsymbol{x}_i)).$$

where $\mathcal{L}$ is a suitable loss function. Here, all nuisance attributes $y^k$ with $k \neq l$ are ignored and we work with samples obtained from the marginal $p(y^l, \boldsymbol{x})$. In practice, however, due to selection bias or other confounding factors in data collection, we are only able to train and test our models on data collected from two related but distinct distributions: $p_{\text{train}}, p_{\text{test}}$. For example, $p_{\text{train}}$ and $p_{\text{test}}$ may be concentrated on different subsets of hospitals and this discrepancy may result in a distribution shift; for example, hospitals may use different equipment, leading to different staining on their cell

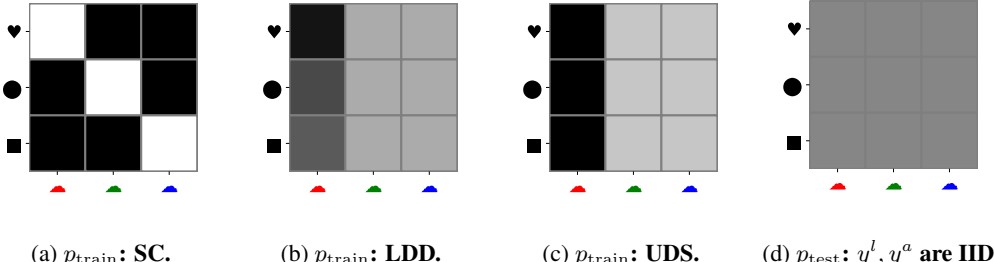

(a) $p_{\text{train}}$: **SC.**      (b) $p_{\text{train}}$: **LDD.**      (c) $p_{\text{train}}$: **UDS.**      (d) $p_{\text{test}}$: $y^l, y^a$ **are IID.**

Figure 1: Visualization of the joint distribution for the different shifts we consider on the DSPRITES example. The lighter the color, the more likely the given sample. figure 1a-1c visualise different shifts over $p_{\text{train}}(y^l, y^a)$ discussed in 2.2: *spurious correlation* (SC), *low-data drift* (LDD), and *unseen data shift* (UDS). figure 1d visualises the test set, where the attributes are uniformly distributed.

images. While we train $f$ on data from $p_{\text{train}}$ by minimizing $\hat{R}(f; p_{\text{train}})$, we aim to learn a model that generalises well to data from $p_{\text{test}}$; that is, it should achieve a small $\hat{R}(f; p_{\text{test}})$.

While generalization in the above sense is desirable for machine learning models, it is not clear *why* a model $f$ trained on data from $p_{\text{train}}$ *should* generalise to $p_{\text{test}}$. It is worth noting that while $p_{\text{train}}$ and $p_{\text{test}}$ can be different, they are both related to the true distribution $p$. We take inspiration from disentanglement literature to express this relationship. In particular, that we can view data as being decomposed into an underlying set of factors of variations. We formalise various distribution shifts using a latent variable model for the true data generation process:

$$z \sim p(z) \qquad\qquad y^i \sim p(y^i|z) \quad i = 1 \ldots K \qquad\qquad \boldsymbol{x} \sim p(\boldsymbol{x}|z) \qquad (1)$$

where $z$ denotes latent factors. By a simple refactorization, we can write

$$p(y^{1:K}, \boldsymbol{x}) = p(y^{1:K}) \int p(\boldsymbol{x}|z)p(z|y^{1:K})dz = p(y^{1:K})p(\boldsymbol{x}|y^{1:K}).$$

Thus, the true distribution can be expressed as the product of the marginal distribution of the attributes with a conditional generative model. We assume that distribution shifts arise when a new marginal distribution for the attributes is chosen, such as $p(y^{1:K}) \neq p_{\text{train}}(y^{1:K}) \neq p_{\text{test}}(y^{1:K})$, but otherwise the conditional generative model is shared across all distributions, i.e., we have $p_{\text{test}}(y^{1:K}, \boldsymbol{x}) = p_{\text{test}}(y^{1:K}) \int p(\boldsymbol{x}|z)p(z|y^{1:K})dz$, and similarly for $p_{\text{train}}$.

To provide more context, as a running example, we use the color DSPRITES dataset (Matthey et al., 2017); where in our notation $y^1$ defines the color with $\mathbb{A}^1 = \{\text{red}, \text{green}, \text{blue}\}$, and $y^2$ defines the shape with $\mathbb{A}^2 = \{\text{ellipse}, \text{heart}, \text{square}\}$. We can imagine that a data collector (intentionally or implicitly) selects some marginal distribution over attributes $p_{\text{train}}(y^{1:K})$ when training; for example they select mostly blue ellipses and red hearts. This induces a new joint distribution over latent factors and attributes: $p_{\text{train}}(z, y^{1:K}) = p(z|y^{1:K})p_{\text{train}}(y^{1:K})$. Consequently, during training, we get images with a different joint distribution $p_{\text{train}}(\boldsymbol{x}, y^{1:K}) = \int p(\boldsymbol{x}|z)p_{\text{train}}(z, y^{1:K})$. This similarly applies when collecting data for the test distribution. We focus on common cases of distribution shifts visualized in figure 1; we discuss these in more detail in section 2.2.

The goal of enforcing robustness to distribution shifts is to maintain performance when the data generating distribution $p_{\text{train}}$ changes. In other words, we would like to minimize risk on $p, p_{\text{test}}$ given *only* access to $p_{\text{train}}$. We can achieve robustness in the following ways:

- **Weighted resampling.** We can resample the training set using importance weights $W(y^{1:K}) = p(y^{1:K})/p_{\text{train}}(y^{1:K})$. This means that given the attributes, the $i$-th data point $(y_i^{1:K}, \boldsymbol{x}_i)$ in the training set is used with probability $W(y_i^{1:K})/\sum_{i'=1}^n W(y_{i'}^{1:K})$ rather than $1/n$. We refer to this empirical distribution as $p_{\text{reweight}}$. This procedure requires access to the true distribution of attributes $p(y^{1:K})$, so to avoid bias and improve fairness, it is often assumed that all combinations of attributes happen uniformly at random.

- **Data Augmentation**: Alternatively, we can learn a generative model $\hat{p}(\boldsymbol{x}|y^{1:K})$ from the training data that aims to approximate $\int p(\boldsymbol{x}|z)p(z|y^{1:K})dz$, as the true conditional

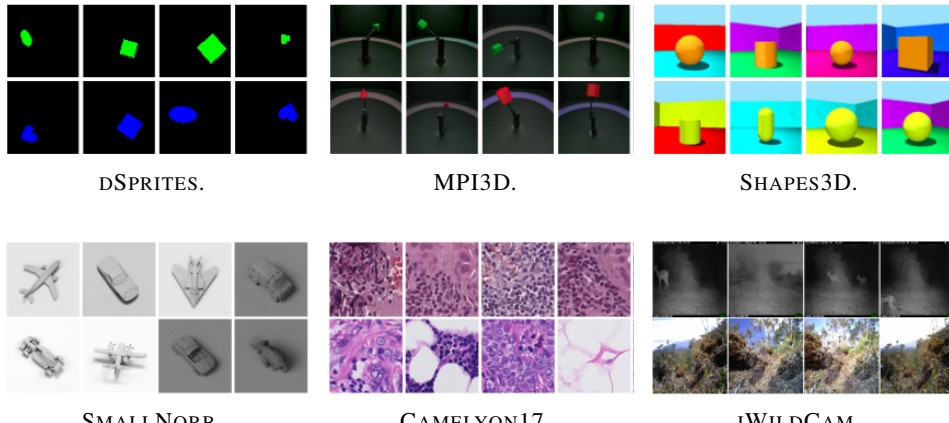

Figure 2: **Dataset samples**. Each row fixes an attribute (e.g. color for DSPRITES, MPI3D, SHAPES3D; azimuth for SMALLNORB; hospital for CAMELYON17; and location for IWILDCAM).

generator is by our assumption the same over all (e.g. train and test) distributions. If such a conditional generative model can be learned, we can sample new synthetic data at training time (e.g. according to the true distribution $p(y^{1:K})$) to correct for the distribution shift. More precisely, we can generate data from the augmented distribution $p_{\text{aug}} = (1 - \alpha)p_{\text{reweight}} + \alpha\hat{p}(\boldsymbol{x}|y^{1:K})p(y^{1:K})$ and train a supervised classifier on this augmented dataset. Here, $\alpha \in [0, 1]$ is the percentage of synthetic data used for training.

- **Representation Learning**: An alternative factorization of a data generating distribution (e.g. train) is $p_{\text{train}}(y^{1:K}, \boldsymbol{x}) = \int p(z|\boldsymbol{x})p_{\text{train}}(y^{1:K}|z)dz$. We can learn an unsupervised representation that approximates $p(z|\boldsymbol{x})$ using the training data only, and attach a classifier to learn a task specific head that approximates $p_{\text{train}}(y^l|z)$. Again, by our assumption $p(z|\boldsymbol{x}) \propto p(\boldsymbol{x}|z)p(z)$. Given a good guess of the true prior, the learned representation would not be impacted by the specific attribute distribution and so generalise to $p_{\text{test}}, p$.

## 2.2 DISTRIBUTION SHIFTS

While distribution shifts can happen in a continuum, we consider three types of shifts, inspired by real-world challenges. We discuss these shifts and two additional, real-world inspired conditions.

**Test distribution** $p_{\text{test}}$**.** We assume that the attributes are distributed uniformly: $p_{\text{test}}(y^{1:K}) = 1/\prod_i |\mathbb{A}^i|$. This is desirable, as all attributes are represented and a-priori independent.

**Shift 1: Spurious correlation – Attributes are correlated under $p_{\text{train}}$ but not $p_{\text{test}}$.** Spurious correlation arises in the wild for a number of reasons including capture bias, environmental factors, and geographical bias (Beery et al., 2018; Torralba & Efros, 2011). These spurious correlations lead to surprising results and poor generalization. Therefore, it is important to be able to build models that are robust to such challenges. In our framework, spurious correlation arises when two attributes $y^a$, $y^b$ are correlated at training time, but this is not true of $p_{\text{test}}$, for which attributes are independent: $p_{\text{train}}(y^a|y^1 \dots y^b \dots y^K) > p_{\text{train}}(y^a|y^1 \dots y^{b-1}, y^{b+1} \dots y^K)$. This is especially problematic when one attribute $y^b$ is $y^l$, the label. Using the running DSPRITES example, shape and color may be correlated and the model may find it easier to predict color. If color is the label, the model will generalise well. However, if the aim is to predict shape, the model's reliance on color will lead to poor generalization.

**Shift 2: Low-data drift – Attribute values are unevenly distributed under $p_{\text{train}}$ but not under $p_{\text{test}}$.** Low-data drift arises in the wild (e.g. in (Buolamwini & Gebru, 2018) for different ethnicities) when data has not been collected uniformly across different attributes. When deploying models in the wild, it is important to be able to reason and have confidence that the final predictions will

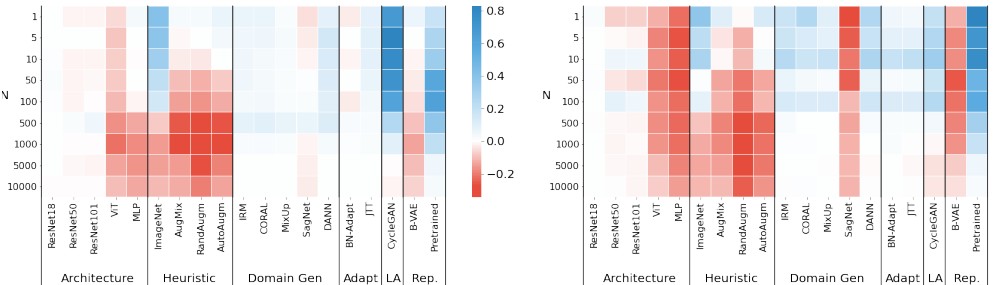

Figure 3: **Spurious Correlation.** We use all correlated samples and vary the number of samples $N$ from the true, uncorrelated distribution. We plot the percentage change over the baseline ResNet, averaged over all seeds and datasets. Blue is better, red worse. CYCLEGAN performs consistently best while ImageNet augmentation and pretraining on ImageNet also consistently boosts performance.

Figure 4: **Low-data drift.** We use all samples from the high data regions and vary the number of samples $N$ from the low-data region. We plot the percentage change over the baseline ResNet, averaged over all seeds and datasets. Blue is better, red worse. Pretraining on ImageNet performs consistently best, while CYCLE-GAN, most domain generalization methods and ImageNet augmentation also provide some boost in performance.

be consistent and fair across different attributes. In the framework above, low-data shifts arise when certain values in the set $\mathbb{A}^a$ of an attribute $y^a$ are sampled with a much smaller probability than in $p_{\text{test}}$: $p_{\text{train}}(y^a = v) \ll p_{\text{test}}(y^a = v)$. Using the DSPRITES example, only a handful of red shapes may be seen at training time, yet in $p_{\text{test}}$ all colors are sampled with equal probability.

**Shift 3: Unseen data shift – Some attribute values are unseen under $p_{\text{train}}$ but are under $p_{\text{test}}$.** This is a special case of *shift 2: low-data drift* which we make explicit due to its important real world applications. Unseen data shift arises when a model, trained in one setting is expected to work in another, disjoint setting. For example: a model trained to classify animals on images at certain times of day should generalise to other times of day. In our framework, *unseen data shift* arises when some values in the set $\mathbb{A}^a$ of an attribute $y^a$ are unseen in $p_{\text{train}}$ but are in $p_{\text{test}}$:

$$p_{\text{train}}(y^a = v) = 0 \qquad p_{\text{test}}(y^a = v) > 0 \qquad |\{v|p_{\text{train}}(y^a = v)\}| > 1 \qquad (2)$$

This is a stronger constraint than in standard out-of-distribution generalization (see section 6), as multiple values for $\mathbb{A}^a$ must be seen under $p_{\text{train}}$, which allows the model to learn invariance to $y^a$. In the DSPRITES example, the color red may be unseen at train time but all colors are in $p_{\text{test}}$.

**Discussion.** We choose these sets of shifts as they are the building blocks of more complex distribution shifts. Consider the simplest case of two attributes: the label and a nuisance attribute. If we consider the marginal distribution of the label, it decomposes into two terms: the conditional probability and the probability of a given attribute value: $p(y^l) = \sum_{y^a} p(y^l|y^a)p(y^a)$. The three shifts we consider control these terms independently: *unseen data shift* and *low-data drift* control $p(y^a)$ whereas *spurious correlation* controls $p(y^l|y^a)$. The composition of these terms describes any distribution shift for these two variables.

## 2.3 CONDITIONS

**Label noise.** We investigate the change in performance due to noisy information. This can arise when there are disagreements and errors among the labellers (e.g. in medical imaging (Castro et al., 2020)). We model this as an observed attribute (e.g. the label) being corrupted by noise. $\hat{y}^i \sim c(y^i)$, where $y^i \in \mathbb{A}^i$ is the true label, $\hat{y}^i \in \mathbb{A}^i$ the corrupted, observed one, and $c$ the corrupting function.

**Dataset size.** We investigate how performance changes with the size of the training dataset. This setting arises when it is unrealistic or expensive to collect additional data (e.g. in medical imaging or in camera trap imagery). Therefore, it is important to understand how performance degrades given fewer total samples. We do this by limiting the total number of samples from $p_{\text{train}}$.

## 3 MODELS EVALUATED

We evaluate 19 algorithms to cover a broad range of approaches that can be used to improve model robustness to distribution shifts and demonstrate how they relate to the three ways to achieve robustness, outlined in section 2. We believe this is the first paper to comprehensively evaluate a large set of different approaches in a variety of settings. These algorithms cover the following areas: architecture choice, data augmentation, domain adaptation, adaptive approaches and representation learning. Further discussion on how these models relate to our robustness framework is in appendix E.

**Architecture choice.** We evaluate the following standard vision models: ResNet18, ResNet50, ResNet101 (He et al., 2016), ViT (Dosovitskiy et al., 2021), and an MLP (Vapnik, 1992). We use weighted resampling $p_{\text{reweight}}$ to oversample from the parts of the distribution that have a lower probability of being sampled from under $p_{\text{train}}$. Performance depends on how robust the learned representation is to distribution shift.

**Heuristic data augmentation.** These approaches attempt to approximate the true underlying generative model $p(\boldsymbol{x}|y^{1:K})$ in order to improve robustness. We analyze the following augmentation methods: standard ImageNet augmentation (He et al., 2016), AugMix without JSD (Hendrycks et al., 2020), RandAugment (Cubuk et al., 2020), and AutoAugment (Cubuk et al., 2019). Performance depends on how well the heuristic augmentations approximate the true generative model.

**Learned data augmentation.** These approaches approximate the true underlying generative model $p(\boldsymbol{x}|y^{1:K})$ by learning augmentations conditioned on the nuisance attribute. The learned augmentations can be used to transform any image $\boldsymbol{x}$ to have a new attribute, while keeping the other attributes fixed. We follow Goel et al. (2020), who use CYCLEGAN (Zhu et al., 2017), but we do not use their SGDRO objective in order to evaluate the performance of learned data augmentation alone. Performance depends on how well the learned augmentations approximate the true generative model.

**Domain generalization.** These approaches aim to recover a representation $z$ that is independent of the attribute: $p(y^a, z) = p(y^a)p(z)$ to allow generalization over that attribute. We evaluate IRM (Arjovsky et al., 2019), DeepCORAL (Sun & Saenko, 2016), domain MixUp (Gulrajani & Lopez-Paz, 2021), DANN (Ganin et al., 2016), and SagNet (Nam et al., 2021). Performance depends on the invariance of the learned representation $z$.

**Adaptive approaches.** These works modify $p_{\text{reweight}}$ dynamically. We evaluate JTT (Liu et al., 2021) and BN-Adapt (Schneider et al., 2020). These methods do not give performance guarantees.

**Representation learning.** These works aim to learn a robust representation of $z$ that describes the true prior. We evaluate using a $\beta$-VAE (Higgins et al., 2017a) and pretraining on ImageNet (Deng et al., 2009). Performance depends on the quality of the learned representation for the specific task.

## 4 EXPERIMENTS

We first introduce the datasets and experimental setup. We evaluate the 19 different methods across these six datasets, three distribution shifts, varying label noise, and dataset size. We plot aggregate results in figures 3-7 and complete results in the appendix in figures 10-12. We discuss the results by distilling them into seven concrete takeaways in section 4.1 and four practical tips in section 4.2.

**Datasets.** We evaluate these approaches on six vision, classification datasets – DSPRITES (Matthey et al., 2017), MPI3D (Gondal et al., 2019), SMALLNORB (LeCun et al., 2004), SHAPES3D (Burgess & Kim, 2018), CAMELYON17 (Koh et al., 2020; Bandi et al., 2018), and IWILDCAM (Koh et al., 2020; Beery et al., 2018). These datasets consist of multiple (potentially an arbitrarily large number) attributes. We select two attributes $y^l, y^a$ for each dataset and make one $y^l$ the label. We then use these two attributes to build the three shifts. Visualizations of samples from the datasets are given in figure 2 and further description in appendix D.1. We discuss precisely how we set up the shifts, choose the attributes, and additional conditions for these datasets in appendix D.2.

**Model selection.** When investigating heuristic data augmentation, domain generalization, learned augmentation, adaptive approaches, and representation learning, we use a ResNet18 for the simpler, synthetic datasets (DSPRITES, MPI3D, SHAPES3D, and SMALLNORB) but a ResNet50 for the

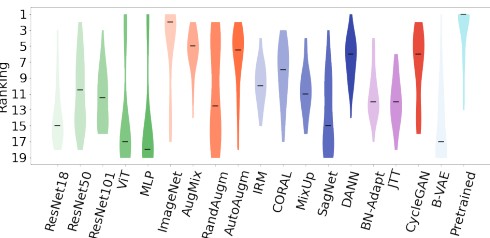

Figure 5: **Unseen data shift.** We rank the methods (where best is 1, worst 19) for each dataset and seed and plot the rankings, with the overall median rank as the black bar. Pretraining on ImageNet and ImageNet augmentation perform consistently best. DANN, CycleGAN and other heuristic augmentations perform consistently well.

more complex, real world ones (CAMELYON17 and IWILDCAM). To perform model selection, we choose the best model according to the validation set which matches the distribution of the test set. In the *unseen data shift* setting for the CAMELYON17 and IWILDCAM, we use the given out-of-distribution validation set, which is a different, distinct set in $\mathcal{D}$ that is independent of $\mathcal{D}_{\text{train}}, \mathcal{D}_{\text{test}}$. (We consider using the in-distribution validation set in appendix B.4.)

**Hyperparameter choices.** We perform a sweep over the hyperparameters (the precise sweeps are given in appendix F.8). We run each set of hyperparameters for five seeds for each setting. To choose the best model for each seed, we perform model selection over *all* hyperparameters using the top-1 accuracy on the validation set. In the *low-data* and *spurious correlation* settings, we choose a different set of samples from the low-data region with each seed. We report the mean and standard deviation over the five seeds.

## 4.1 TAKEAWAYS

**Takeaway 1: While we can improve over ERM, no one method always performs best.** The relative performance between methods varies across datasets and shifts. Under *spurious correlation* (figure 3), CYCLEGAN consistently performs best but in figure 4, under *low-data drift*, pretraining consistently performs best. Under *unseen data shift* (figure 5), pretraining is again one of the best performing models. However, if we drill down on the results in figure 10 (appendix B.1), we can see pretraining performs best on the synthetic datasets, but not on CAMELYON17 (where using augmentation or DANN is best) or IWILDCAM (where using ViT or an MLP is best).

**Takeaway 2: Pretraining is a powerful tool across different shifts and datasets.** While pretraining is not always helpful (e.g. in appendix B.1 on CAMELYON17 in figures 10-11, IWILDCAM in figures 10-11), it often provides a strong boost in performance. This is presumably because the representation $z$ learned during pretraining is helpful for the downstream task. For example, the representation may have been trained to be invariant to certain useful properties (e.g. scale, shift, and color). If these properties are useful on the downstream tasks, then the learned representation should improve generalization.

**Takeaway 3: Heuristic augmentation improves generalization *if* the augmentation describes an attribute.** In all settings (figures 3-5), ImageNet augmentation generally improves performance. However, RandAugment, AugMix, and AutoAugment have more variable performance (as further shown in figures 10-12). These methods are compositions of different augmentations. We investigate the impact of each augmentation in RandAugment in appendix B.2 and find variable performance. Augmentations that approximate the true underlying generative model $p(\boldsymbol{x}|y^{1:K})$ lead to the best results; otherwise, the model may waste capacity. For example, on CAMELYON17 (which consists of cell images), color jitter harms performance but on SHAPES3D and MPI3D it is essential.

**Takeaway 4: Learned data augmentation is effective across different conditions and distribution shifts.** This approach is highly effective in the *spurious correlation* setting (figure 3). It can also help in the *low-data* and *unseen data shift* settings (figure 4,5) (though the gains for these two shifts are not as large as for pretraining). The effectiveness of this approach can be explained by the fact that if the augmentations are learned perfectly, then augmented samples by design are from the true underlying generative model and can cover missing parts of the distribution.

**Takeaway 5: Domain generalization algorithms offer limited performance improvement.** In some cases these methods (in particular DANN) do improve performance, most notably in the *low-data drift* and *unseen data shift* settings (figures 4-5). However, this depends on the dataset (see figures 10-12) and performance is rarely much better than using heuristic augmentation.

Figure 6: **Condition 1: Noisy labels.** We vary the amount of noise $p$ in the labels. We plot the percentage change over the baseline ResNet, averaged over all seeds and datasets.

Figure 7: **Condition 2: Fixed data.** We vary the total size of the dataset $T$. We plot the percentage change over the baseline ResNet, averaged over all seeds and datasets.

**Takeaway 6: The best algorithms may differ under the precise conditions.** When labels have varying noise in figure 6, relative performance is reasonably consistent. When the dataset size decreases in figure 7, heuristic augmentation methods perform poorly. However, using pretraining and learned augmentation is consistently robust.

**Takeaway 7: The precise attributes we consider directly impacts the results.** For example, on DSPRITES, if we make color $y^l$ and shape $y^a$, we find that *all* methods generalise perfectly in the *unseen data shift* setting (as demonstrated in appendix B.3) unlike when shape is $y^l$ (figure 10).

## 4.2 PRACTICAL TIPS

While there is no free lunch in terms of the method to choose, we recommend the following tips.

**Tip 1: If heuristic augmentations approximate part of the true underlying generative model, use them.** Under this constraint, heuristic augmentations can significantly improve performance; this should be a first point of call. How to heuristically choose these augmentations without exhaustively trying all possible combinations is an open research question.

**Tip 2: If heuristic augmentations do not help, learn the augmentation.** If the true underlying generative model cannot be readily approximated with heuristic techniques, but some subset of the generative model can be learned by conditioning on known attributes, this is a promising way to further improve performance. How to learn the underlying generative model directly from data and use this for augmentation is a promising area to explore more thoroughly.

**Tip 3: Use pretraining.** In general, pretraining was found to be a useful way to learn a robust representation. While this was not true for all datasets (e.g. CAMELYON17, IWILDCAM), performance could be dramatically improved by pretraining (DSPRITES, MPI3D, SMALLNORB, SHAPES3D). An area to be investigated is the utility of self-supervised pre-training.

**Tip 4: More complex approaches lead to limited improvements.** Domain generalization, adaptive approaches and disentangling lead to limited improvements, if any, across the different datasets and shifts. Of these approaches, DANN performs generally best. How to make these approaches generically useful for robustness is still an open research question.

## 5 DISCUSSION

Our experiments demonstrate that no one method performs best over all shifts and that performance is dependent on the precise attribute being considered. This leads to the following considerations.

**There is no way to decide a-priori on the best method given only the dataset.** It would be helpful for practitioners to be able to select the best approaches without requiring comprehensive evaluations and comparisons. Moreover, it is unclear how to pinpoint the precise distribution shift (and thereby methods to explore) in a given application. This should be an important future area of investigation.

**We should focus on the cases where we have knowledge about the distribution shift.** We found that the ability of a given algorithm to generalize depends heavily on the attribute and dataset being

considered. Instead of trying to make one algorithm for any possible shift, it makes sense to have adaptable algorithms which can use auxiliary information if given. Moreover, algorithms should be evaluated in the context for which we will use them.

**It is pivotal to evaluate methods in a variety of conditions.** Performance varies due to the number of examples, amount of noise, and size of the dataset. Thus it is important to perform comprehensive evaluations when comparing different methods, as in our framework. This gives others a more realistic view of different models' relative performance in practice.

## 6 RELATED WORK

We briefly summarize benchmarks on distribution shift, leaving a complete review to appendix C.

**Benchmarking robustness to out of distribution (OOD) generalization.** While a multitude of methods exist that report improved OOD generalization, Gulrajani & Lopez-Paz (2021) found that in actuality no evaluated method performed significantly better than a strong ERM baseline on a variety of datasets. However, Hendrycks et al. (2021) found that, when we focus on better augmentation, larger models and pretraining, we can get a sizeable boost in performance. This can be seen on the Koh et al. (2020) benchmark (the largest boosts come from larger models and better augmentation). Our work is complementary to these methods, as we look at a range of approaches (pretraining, heuristic augmentation, learned augmentation, domain generalisation, adaptive, disentangled representations) on a range of both synthetic and real-world datasets. Moreover, we allow for a fine-grained analysis of methods over different distribution shifts.

**Benchmarking spurious correlation and low-data drift.** Studies on fairness and bias (surveyed by Mehrabi et al. (2021)) have demonstrated the pernicious impact of low-data in face recognition (Buolamwini & Gebru, 2018), medical imaging (Castro et al., 2020), and conservation (Beery et al., 2018) and spurious correlation in classification (Geirhos et al., 2019) and conservation (Beery et al., 2020). Arjovsky et al. (2019) hypothesized that spurious correlation may be the underlying reason for poor generalization of models to unseen data. To our knowledge, there has been no large scale work focused on understanding the benefits of different methods across these distribution shifts systematically across multiple datasets and with fine-grained control on the amount of shift. Here we introduce a framework for creating these shifts in a controllable way to allow such challenges to be investigated robustly.

**Benchmarking disentangled representations.** A related area, disentangled representation learning, aims to learn a representation where the factors of variation in the data are separated. If this could be achieved, then models should be able to generalise effortlessly to unseen data as investigated in multiple settings such as reinforcement learning (Higgins et al., 2017b). Despite many years of work in disentangled representations (Higgins et al., 2017a; Burgess et al., 2017; Kim & Mnih, 2018; Chen et al., 2018), a benchmark study by Locatello et al. (2019) found that, without supervision or implicit model or data assumptions, one cannot reliably perform disentanglement; however, weak supervision appears sufficient to do so (Locatello et al., 2020). Dittadi et al. (2021); Schott et al. (2021); Montero et al. (2020) further investigated whether representations (disentangled or not) can interpolate, extrapolate, or compose properties; they found that when considering complex combinations of properties and multiple datasets, representations do not do so reliably.

## 7 CONCLUSIONS

This work has put forward a general, comprehensive framework to reason about distribution shifts. We analyzed 19 different methods, spanning a range of techniques, over three distribution shifts – *spurious correlation*, *low-data drift*, and *unseen data shift*, and two additional conditions – *label noise* and *dataset size*. We found that while results are not consistent across datasets and methods, a number of methods do better than an ERM baseline in some settings. We then put forward a number of practical tips, promising directions, and open research questions. We hope that our framework and comprehensive benchmark spurs research on in this area and provides a useful tool for practitioners to evaluate which methods work best under which conditions and shifts.

ACKNOWLEDGMENTS

The authors thank Irina Higgins and Timothy Mann for feedback and discussions while developing their work. They also thank Irina, Rosemary Ke, and Dilan Gorur for reviewing earlier drafts.

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
