# OpenReview forum: "A Fine-Grained Analysis on Distribution Shift"
_ICLR.cc/2022/Conference — ICLR 2022 Oral_

### Official Review · Reviewer_zAfw · 2021-10-25

**Correctness:** 2
**Technical Novelty And Significance:** 2
**Empirical Novelty And Significance:** 3
**Recommendation:** 8
**Confidence:** 4

**Main Review:**

===================
Strengths:
In my opinion, the efforts to unify all the studies of distribution shifts into one general framework is the greatest contribution made by this paper. Recently, plenty of distribution shift settings become popular in various machine learning fields, such as long-tailed classification/detection, domain adaptation, out-of-distribution generalization, etc. However, they are all studied independently, despite the fact that a similar essence is shared by them under the surface. This framework puts them into the same testbed, which allows us to evaluate the scope and limitation of different methods in all kinds of settings and data types. Besides, the authors also provided a number of practical tips and useful conclusions based on massive experiments under diverse settings and datasets.

===================
Weaknesses:
- However, some conclusions in this paper violate my own observations. For example, the heuristic data augmentation method, Rand Augmentation, is found to be very helpful in long-tailed classification (a specific type of low-data drift) and some label noise tasks. It can consistently improve 2-5 points of accuracy across different datasets and settings. Yet, Rand Augmentation looks to be the worst method in Figures 3 and 4. One possible explanation is that datasets I was using are all from real-world images, so the Rand Aug can reveal the underlying generative model p(x|y1:K), while the datasets used in this paper are mostly synthetic images or medical images, where Rand Aug didn't approximate the true underlying generative model p(x|y1:K). In order to prevent the conclusions in this paper from misleading future researchers, I suggest the authors summarize datasets into different types (like synthetic dataset, medical dataset, and normal dataset) and investigate them separately.
- Another concern is that should the ImageNet pretraining be used as a valid method for the study of robustness to distribution shifts? It may violate the settings of SC, LDD, and UDS by introducing samples from unknown distributions. For example, unseen data distribution in the given task and dataset may be compensated by the ImageNet dataset. That's why pretraining is usually forbidden in the tasks like OOD image classification or long-tailed classification. As a result, the superior performance of pertaining is not just frustrating but also (could be) unfair.

**Summary Of The Paper:**

===================
Summary:
The robustness to distribution shifts is one of the biggest concerns in deploying machine learning systems. This paper summarizes all distribution shifts into three prototypes: spurious correlation, low-data drift, and unseen data shift, together with two additional conditions: label noise and dataset size. Other more complex distribution shifts can be regarded as compositions of these components. After that, they evaluated19 methods, spanning 5 categories: architecture choice, data augmentation, domain generalization, adaptive algorithms, and representation learning, in 6 datasets. In the end, they provided a number of practical tips, useful conclusions, and promising directions for future researchers in this field.

**Summary Of The Review:**

===================
Justification:
I recognize the value of the proposed comprehensive framework and all systematic studies made by this paper. I would be more than willing to accept the paper once my concerns are properly addressed.

---

> ### Author Response · Authors · 2021-11-16
> **Response to reviewer zAfw**
>
> We thank the reviewer for appreciating our efforts and for their thoughtful comments.
>
> *Question 1: However, some conclusions in this paper violate my own observations. For example, the heuristic data augmentation method, Rand Augmentation, is found to be very helpful in long-tailed classification (a specific type of low-data drift) and some label noise tasks.*
>
> We thank the reviewer for pointing this out. We noticed when running the RandAugment experiments that some augmentations were more helpful than others. As a result, we explored the impact of each augmentation for each dataset in the appendix in figure 8. We found that the utility of each augmentation depended on whether that augmentation promoted invariance to some nuisance attribute in the underlying generative model.  For example, on MPI3D, color augmentations (color, contrast, brightness, etc) improve performance the most whereas spatial transformations (shear, translate, and rotate) hurt performance. On MPI3D, objects are always placed in the same position in the image and there is no spatial transformation, whereas there are variations in color. We highlighted and discussed this point in one of the takeaway messages: **takeaway 3: Heuristic augmentation does not always improve generalization**.  We’ve also included a couple of sentences in takeaway 3 discussing this point, and have referenced a relevant recent work that explores this in depth in the context of common corruptions [1].
>
> To clarify further that performance varies over datasets, as suggested, we have added a couple of lines in the dataset description (section 4, paragraph **Datasets**): “These datasets differ in their properties: dSprites, MPI3D, Shapes3D are all synthetic; SmallNorb consists of black and white images of toys; Camelyon consists of medical imagery and iWildCam of camera trap imagery. While we aggregate results in the main text, results differ across datasets so please refer to figures 10-12 for performance on each dataset.”
>
> *Question 2: Another concern is that should the ImageNet pretraining be used as a valid method for the study of robustness to distribution shifts? It may violate the settings of SC, LDD, and UDS by introducing samples from unknown distributions. For example, unseen data distribution in the given task and dataset may be compensated by the ImageNet dataset. That's why pretraining is usually forbidden in the tasks like OOD image classification or long-tailed classification. As a result, the superior performance of pertaining is not just frustrating but also (could be) unfair.*
>
> We agree with the reviewer’s point that pretraining gives, effectively, the model more data to train with, which can be unfair with respect to other methods. We include this approach, as it is an extremely common method to improve model performance in practice, and so we believe it is valuable to understand how it performs in comparison to other methods. We note that while it is unfair (in that ImageNet pretraining effectively uses more data), ImageNet pretraining actually does not always perform particularly better than the baseline and is sometimes outperformed by other methods (e.g. on iWildCam and Camelyon in figures 10-11).
> We leave it to people using our framework to decide on the most suitable set of approaches when performing their own studies (for example all methods can be run with pretraining initially or it can be held out). In the text, we have clarified that pretraining means that the model has been trained with additional data (see section 4, paragraph **additional data**).
>
> [1] Eric Mintun et al. “On Interaction Between Augmentations and Corruptions in Natural Corruption Robustness”

---

> > ### Comment · Reviewer_zAfw · 2021-11-18
> > **Re: Response to reviewer zAfw**
> >
> > Thank you for your replies. I've raised my final score to 8 based on your response. Although the early version of the paper missed some important discussions as I mentioned, I admire the efforts and the elegant formulation to unify those distribution shift tasks. Hope that the final version of the paper could bring more insights to the audience in the corresponding fields.

---

### Official Review · Reviewer_U5qL · 2021-10-28

**Correctness:** 4
**Technical Novelty And Significance:** 3
**Empirical Novelty And Significance:** 3
**Recommendation:** 10
**Confidence:** 4

**Main Review:**

“find that a model trained on one set of hospitals may not generalise to the imaging conditions of another.” -> Training on “one set of hospitals” is a weird formulation and it’s not clear what is meant; set of MRIs from multiple hospitals or similar? Please rewrite this sentence.


“These methods span the following 5 common approaches: architecture choice, data augmentation, domain generalization, adaptive algorithms, and representation learning.” The collection of approaches written in this way sounds a bit strange to me because the choices are neither orthogonal nor of the same kind. Architecture choice is a property of the model being trained (irrespective of the task), data augmentation heavily depends on the task, domain generalization IS a task, and representation learning is a whole area of machine learning. At this stage of the paper, I find this list very confusing and would suggest rewriting or clarifying this sentence.


“We assume that distribution shifts arise…” Distribution (or covariate) shift is usually defined using the definition in [1], section 2.1.1. Can you please comment on whether there is difference to your definition and what it means for the approaches one would use to tackle it? From my understanding, it is actually the same because in [1], covariate shift is defined as (a) P(X) != P’(X) and (b) P (Y |X) = P’(Y |X). Here, the authors also have the condition (b) and write that P(y_1:k) != p_train(y_1:k) != p_test(y_1:k) which will result in the condition (a) P(x) != P’(X). In any case, distribution shift is a well-defined phenomenon; thus, it would be good if the authors could cite some standard definition (e.g., [1]) and comment why their definition is different (if it is different).


Page 3: Data Augmentation. The authors write “alternatively” and I believe they refer to the previously proposed reweighting method. Writing “alternatively” means to me that they describe the methods separately first. Therefore, I am confused by the definition of p_aug since it still contains p_reweight. The authors should either (1) remove p_reweight from p_aug or (2) write “additionally” instead of “alternatively”. I would prefer (1).


Page 3: Data Augmentation. I don’t quite agree with how data augmentation is defined here. The authors write that one can synthesize artificial data with a generative model that aims to approximate the true generative model and use this data as data augmentation. I don’t think this understanding of using data augmentation is how researchers generally think about data augmentation. I think the main driving factor for using data augmentation is to artificially increase the dataset size and to make the classifier more smooth around the data points, i.e. its decision should not change if the data distribution changes slightly. For this, researchers generally do not try to learn the true generative model, but rather use simple augmentations such as e.g., Gaussian noise, crops and horizontal flips. Later, the authors write that they use AugMix, RandAugment and AutoAugment as augmentations. All these augmentations are simple parametric functions that neither depend on the latent z nor on x since they can be used irrespective of the data and/or task. Learning the generative model is more of a GAN-like approach, so I would maybe split the paragraph into something like (1) Standard/Simple/Parametric/Heuristic data augmentation and (2) Learned data augmentation. The authors actually split the two notions of data augmentation in section 3. I would suggest just using the same split here. The authors actually claim in Takeaway 3 that “Heuristic augmentation improves generalization if the augmentation describes an attribute.” Nevertheless, I think defining data augmentation in this way here is a bit preemptive.

Page 7: “We report the mean and standard deviation over the five seeds.” I absolutely appreciate the effort done here for reporting the error over 5 runs as this is unfortunately not standard over even common in most papers.


Page 4: “Test distribution…” please define A_i


Page 4: “Shift 1:” Should it be P_test on the right hand side of the definition?


Page 4: “Shift 2: Low-data drift” -> Maybe mentioning that this issue is being studied by the fairness community would be good.


Page 4: “Shift 3: Unseen data shift – Some attribute values are unseen under ptrain but are under ptest.” Suggest to add “present” as in “but are present under p_test”.


Shift 3: “which we make explicit due to its important real world applications” -> importance


Shift 3: I do not understand the right-most inequality in Eq.2. Please explain.


I really like the structure and content of section 2.2. The authors put a lot of effort into properly and carefully defining and disentangling different distribution shifts. I also appreciate the provided examples from DSprites to illustrate all the different shifts.


Figure 3: I find it weird that using more samples from the true distribution seems to hurt performance in many cases. Additionally, checking Figure 11 in the Appendix, it looks like higher N is correlated with higher accuracy though it is hard to judge only looking at the image. The authors should comment on this.


Figure 4: Does this Figure show the accuracy on the biased or non-biased datasets? Similar to Figure 11, in Figure 12, it again looks like a higher N results in higher accuracy. In Figure 12, some bars are missing.


Figures 10, 11 and 12 are generally hard to read because there are so many bars and the only information one can extract is that there is a monotonic increase over the different models which is pretty useless since the ordering has likely been chosen such that there is a monotonic increase. Judging how N changes the bar heights is not really possible. I would suggest to maybe plot these Figures as heat plots similar to Figures 3 and 4. With heat plots, the authors could also write the resulting accuracy numbers into the heat map squares. This will allow researchers to cite their numbers in the future and also make reproducibility easier since these numbers can be easily compared against.


It is annoying that the Figures 10-12 in the Appendix are not plotted together and also not close to the corresponding text. I would highly suggest restructuring the Appendix to make parsing it easier, especially, since it is very long. For example, the code for the framework can be put at the very end such that it does not break the flow when looking at the additional results.


Takeaway 3. I do not see how the presented evidence leads to the drawn conclusion that “Heuristic augmentation improves generalization if the augmentation describes an attribute.” In B.2, the authors merely show that “No augmentation always leads to a strong boost in performance.” But it is not discussed in what way the successful augmentations approximate the true generative model. This type of analysis has done on ImageNet-C and CIFAR10-C in ref. [2] which the authors should cite here. In ref. [2], the authors define a minimal sample distance between the expected distribution shift and the added data augmentation and find strong correlation between the two. Here, a similar analysis would need to be performed in order to be able to do such a claim. In this light, tip 1 is also not grounded on evidence (although it is pretty obvious).


I did not understand whether model selection is part of the framework code?


The tips in 4.2. are not really surprising (or novel). In my opinion, tip 1 is not grounded on the presented evidence since the authors did not show results that would support their takeaway 3. It is an obvious tip though and intuitively, it makes absolute sense, but the authors did not show results to support it. Considering tip 2, I think learning the perfect generative model is a hard task and I am not sure how feasible this task is for ImageNet scale datasets. Given how difficult and unstable GAN training can be, I am not sure how practical this tip is. The authors also did not show results on ImageNet, so it is hard to judge whether this tip would scale to ImageNet. Considering tip3, it is well known that pretraining on ImageNet is powerful as is has been shown in numerous previous works. As for tip4, I am not aware that DANN has been scaled to ImageNet. Additionally, DANN training (being a minimax optimization problem) can be unstable and depend on hyperparameters. Thus, I question the practicality of this tip.
	Some of these tips are not novel and the authors (merely) provide additional evidence for them. It would be nice if the authors could cite some papers where these findings have also been reported.


References:
[1] Bernhard Schölkopf et al. “On causal and anticausal learning”.
[2] Eric Mintun et al. “On Interaction Between Augmentations and Corruptions in Natural Corruption Robustness”




**Summary Of The Paper:**

The authors perform an extensive study of different types of distribution shifts to judge which methods perform best on which dataset and/or method. For this, they carefully define which distribution shifts they are interested in and perform a large scale study for each of the shifts. They propose a standardized framework (which can easily be extended to new methods) to train and evaluate new models and methods on the distribution shifts.

**Summary Of The Review:**

I find the paper well written and easy to follow. The studied distribution shifts are thoughtfully and carefully defined. The authors perform a ton of experiments for the distribution shifts they want to study. I think the proposed benchmark / framework is a great addition to the research community as it will standardize model training and evaluation.


Points I would especially like to see addressed during the rebuttal (partially copied the most important points to me from the main review):

1.	Figure 3: I find it weird that using more samples from the true distribution seems to hurt performance in many cases. Additionally, checking Figure 11 in the Appendix, it looks like higher N is correlated with higher accuracy though it is hard to judge only looking at the image. The authors should comment on this.

2.	Figure 4: Does this Figure show the accuracy on the biased or non-biased datasets? Similar to Figure 11, in Figure 12, it again looks like a higher N results in higher accuracy. In Figure 12, some bars are missing.

3.	Takeaway 3. I do not see how the presented evidence leads to the drawn conclusion that “Heuristic augmentation improves generalization if the augmentation describes an attribute.” In B.2, the authors merely show that “No augmentation always leads to a strong boost in performance.” But it is not discussed in what way the successful augmentations approximate the true generative model. This type of analysis has been done on ImageNet-C and CIFAR10-C in ref. [2] which the authors should cite here. In ref. [2], the authors define a minimal sample distance between the expected distribution shift and the used data augmentation and find strong correlation between the two. Here, a similar analysis would need to be performed in order to be able to do such a claim. In this light, tip 1 is also not grounded on evidence (although it is pretty obvious).

4. Contribution 1: "We propose a framework to define when and why we expect methods to generalise." I think the authors addressed the "when" question with their benchmark, but not really the "why" question. Can the authors comment on which results lead them to conclude why a certain method should generalize?


-> I am also happy to discuss any other points I mentioned but these would be the most important ones to me.

---

> ### Author Response · Authors · 2021-11-16
> **Rebuttal of Main Questions raised by Reviewer U5ql**
>
> We thank the reviewer for their thorough and insightful review. We have made the suggested stylistic changes and appendix changes and have answered questions below. In this comment we focus on the main points raised by the reviewer. We discuss additional points in the second comment.
>
> - *Question 1 and 2: More samples from the true distribution seem to hurt performance.*
>
> We think there was some confusion here. Higher N **does** lead to higher accuracy (as shown in Figure 11, Figure 12). In Figure 3, 4 (as stated in the captions) we are showing the average percent improvement **over** the ResNet baseline on the IID test set (so the non-biased dataset). As a result, what figures 3 and 4 show is that as N gets higher, there is less improvement over the ResNet baseline. We did it this way in order to average over the different datasets.
>
> To clarify that the results are aggregate and on the test set, we have updated text in the first paragraph in section 4 to include: “We plot aggregate results in figures 3-7 by averaging results over the different datasets and complete results in the appendix in  figures 10-12, in which the results are broken down by dataset. All results reported are on the test set.”
>
> We are not sure which bars the reviewer is referring to when they say there are bars missing in Figure 12. If the reviewer could point out the missing method, we will update the plot.
>
> - *Question 3: Takeaway 3.*
>
> This was concluded based on the augmentations that performed best for the different datasets. For example, on MPI3D, in general color augmentations (color, contrast, brightness, etc) improve performance the most whereas spatial transformations (shear, translate, and rotate) hurt performance. On MPI3D, objects are always placed in the same position in the image and there is no spatial transformation, whereas there are variations in color. Therefore, augmentations that promote invariance to color transformations improve performance. Based on a similar analysis we concluded the statement. In takeaway 3, we have included a reference to [2], some more sentences discussing which augmentations help and don’t, and have reworded the takeaway to: “ Heuristic augmentation does not always improve generalization.”.
>
> In comparison to [2], we note that setting up such an analysis is not so simple. We cannot simply compare the two distributions (as in [2]). Consider the simple example of color changes. An augmentation that inverts may be preferable to one that increases sharpness, as inversion decorrelates the precise color from the prediction. However, in perceptual or L2 space, the sharper image will be more similar to the original. Therefore, augmentations that promote invariance to properties of the true underlying generative model are preferable. The most straightforward way to evaluate this is to train a classifier on these augmentations and evaluate whether this improves downstream performance, as we did.
>
> - *Question 4: Question about contribution 1.*
>
> With the ‘when’ and ‘why’ statements we were referring to section 2 which defines how shifts occur and how different general approaches (e.g. **weighted resampling**, **heuristic data augmentation**, **learned data augmentation**, and **representation learning**) can be used to improve robustness to these methods. However, as this was unclear, we have modified this sentence to say: “We propose a framework to unify different distribution shifts, defining how they arise and how different common approaches promote robustness to these shifts.”

---

> > ### Comment · Reviewer_U5qL · 2021-11-19
> > **Response to the authors' comments**
> >
> > I thank the authors for addressing my comments. I read through their responses and the changes in the revised paper version. I will respond to the ones I flagged as the most important ones in my initial review, please consider the ones I do not respond to as addressed adequately. Could the authors maybe publish a csv file with the full results of all models, such that interested researchers can quickly check the numbers? I think that the main benefit of the paper is the standardized benchmark. If the authors can release all numbers, it would make it easy for other researchers to compare. My score has been 8, but I would raise to a 10 if the authors released this csv file, because releasing the accuracy numbers would strengthen the benchmark and make it more useable for other researchers. For example, the timm repository which contains most relevant pytorch models (with pretrained checkpoints) has a csv reporting the accuracy numbers on ImageNet, including relevant preprocessing hyperparameters: https://github.com/rwightman/pytorch-image-models/blob/master/results/results-imagenet.csv. Maybe the authors could post something similar, alongside with relevant hyperparameter choices.
> >
> > ### Question 1 and 2: More samples from the true distribution seem to hurt performance.
> > Yes, thank you for the clarification, it makes sense now.
> >
> > ### Missing bars, Figs. 10-12.
> > Considering the missing bars, I think my confusion stemmed from the change in ordering of the bars. But the bars did not disappear but merely changed their position. I still find this ordering makes it hard to read the bar plots in Figs. 10-12, but I understand that it is a trade-off of what one wants to show. I understand that seeing small differences fast would be harder with a heat map. I still think that heat maps would be better, but as the other reviewers do not seem to share my concerns, I am fine with this display.
> >
> > ### Question 3: Takeaway 3.
> > I appreciate the authors commenting on this question. And I agree with their response that an analysis similar to [2] would not work here. In contrast to ImageNet-C for example, we not have the mapping x_target = x_source+corruption and therefore, calculating the perceptual distance between x_corrupted and x_augmented is impossible.

---

> > > ### Author Response · Authors · 2021-11-19
> > > **Response to Reviewer U5qL's 2nd Response**
> > >
> > > *Could the authors maybe publish a csv file with the full results of all models, such that interested researchers can quickly check the numbers?*
> > >
> > > We thank the reviewer for appreciating our work and will release a CSV with the full set of results with the code release.
> > >
> > > In the meantime, we have added to the supplementary an xlsx spreadsheet containing results for the three distribution shifts (**unseen data shift**, **low data drift** and **spurious correlation**). We include the best result for each seed for each model (over the hyperparameter sweep for that seed and model).
> > > We do this for each value of $N$ for the **low data drift** and **spurious correlation** settings. We also include the results for Figure 9 and the results using the in distribution or out-of-distribution validation set (Figure 10).

---

> > > > ### Comment · Reviewer_U5qL · 2021-11-19
> > > > **Response to the csv file**
> > > >
> > > > Thanks, awesome work! I have raised my score to 10. For MPI3D, is is the accuracy in D-31? Could you please add a short description of the csv later?

---

> > > > > ### Author Response · Authors · 2021-11-19
> > > > > **Response to question on CSV file**
> > > > >
> > > > > Yes it is the accuracy in D-31. Yes we will add a description to the CSV, explaining the organization of results.

---

> ### Author Response · Authors · 2021-11-16
> **Rebuttal of Additional Comments raised by Reviewer U5ql**
>
> Here we address additional comments raised by the reviewer, not covered in the main questions.
>
> - *Distribution (or covariate) shift is usually defined using the definition in [1], section 2.1.1. Can you please comment on whether there is difference to your definition and what it means for the approaches one would use to tackle it?*
>
> The definition of covariate shift as pointed out by the reviewer is a generic one where directly the distribution of observables $p(X)$ is changing, while the conditional distribution of labels, given inputs $p(Y|X)$ is invariant. While this is a mathematically consistent and well defined model, this approach just says that there is a (potentially arbitrary) shift present in $p(X)$ but it does not model how this shift is happening. In contrast, in our approach we assume that actually the underlying generative process $p(X|Z)$, as well as the labelling process $p(Y|Z)$ are invariant; but the distribution shift is happening due to a shift in the marginal distribution chosen for the attributes $p_\textrm{new}(Y)$. In our opinion, this is the more realistic scenario -- for example, when collecting a dataset of medical images, the actual physical process of imaging $p(X|Z)$ would be invariant (as dictated by the laws of physics) while we may have control over the attributes such as age, gender, ethnicity of the patients, or the brand of the imaging equipment, particular hospital where the images are taken etc. The particular choice of the frequency of attributes induces a shift in the prior distribution of the latents from $p(Z) = \sum_Y p(Z| Y) p_\textrm{true}(Y)$ to $p_\textrm{new}(Z) = \sum_Y p(Z| Y) p_\textrm{new}(Y)$. If we integrate over $Z$ and sum over $Y$, we get  $p_\textrm{new}(X)$ that is in general different from $p(X)$ so we have a distribution shift.
>
> We have added the following in Section 2.1 to clarify this: “This is a special case ... may vary.”.
>
> - *Data Augmentation. I don’t quite agree with the definition here.*
>
> We appreciate the point that learning the generative model is not the practitioner's aim when utilising heuristic augmentations. Our aim here is to give a different interpretation of why these augmentations can be useful. We can view heuristic augmentation as attempting to create samples according to some known set of invariances within the domain (e.g. the model should be invariant to color changes). We discuss this further in our answer to question 4 in the main questions. If we could construct augmentations to describe all invariances, we could cover the true generative model. We have clarified this in the paper, by adding an additional paragraph **heuristic data augmentation** in Section 2.1.
>
> - *The tips in 4.2. are not really surprising.*
>
> While the tips are not necessarily surprising, it is good to validate that they are nevertheless true on these distribution shifts and when evaluated comprehensively. We have improved the grounding and wording as follows. For tip 1, we have improved the takeaway as stated in Answer 14 and have used this to ground our tip. For tip 2, we have added the qualitative sentence that this depends on the complexity of the data and quality of the generative model learned. For tip 3, we have included multiple references to previous work. Finally, for tip 4 we have removed the comment about DANN.
>
> - *I do not understand the right-most inequality in Eq.2*
>
> It means the number of values that we see for the nuisance attribute is $> 1$ (e.g. we see at least two different colours when generalising to a third). We have updated the sentence describing how this shift manifests on dSprites to make this explicit: “In the dSprites example, ...”.
>
> - *What about heatmaps in Figures 10-13?*
>
> We did think about this. However, we found that it made distinguishing small differences in model performance hard. Instead, we color code the bars and sort them. The color of the bar denotes the method, colored by type so a reader can extract which method does best as well as which group of methods.
>
> - *I did not understand whether model selection is part of the framework code?*
>
> The framework allows for the metric and validation set used for model selection to be specified. We use this in Figure 10 in order to compare using an OOD versus ID validation set.
>
> - *please define $A_i$*
>
> See section 2.1 in the first sentence: “We assume a joint distribution…”.
>
> - *Should it be $p_\textrm{test}$ on the right hand side of the definition?*
>
> No. All we’re saying is that the probability of a given attribute at train is correlated with another (as opposed to in $p_\textrm{test}$ where the attributes are independent).
>
> - *“Shift 2: Low-data drift” -> Maybe mentioning that this issue is being studied by the fairness community would be good.*
>
> We discuss work on fairness and bias in section 6 (paragraph benchmarking spurious correlation and low data drift). If there are other specific references the reviewer thinks are missing, we are happy to update the paper.

---

### Official Review · Reviewer_HwsU · 2021-11-01

**Correctness:** 3
**Technical Novelty And Significance:** 4
**Empirical Novelty And Significance:** 4
**Recommendation:** 8
**Confidence:** 4

**Main Review:**

### Strengths
-----------------------------

- Firstly, this paper is a job well done! It was written very clearly and most parts were very easy to follow. The motivations of the paper are very clear. The major strength of the paper lies in the extensive experimental evaluation provided across the main paper and the supplementary, which may be used as a standardized reference for many future works.

- The efforts made towards stringing various works aimed at generalization, as well as various natures of distribution shifts, on a common thread is a commendable effort.

- The figures and the graphical illustrations are very well made, and effectively distill the major inferences of the experiments.

### Required Clarifications
-----------------------------

- My major question to the authors of the paper is how general the proposed framework? You seem to select different datasets, but take only two attributes on each datasets which raises questions if the findings of the study hold with larger scale real world datasets with multiple factors of variation, often heavily entangled. I understand that these settings are chosen to keep the experiments tractable, but it would be helpful to provide insight into whether the framework holds for Imagenet scale datasets with unknown (and possibly unobservable) factors of variation [1].
For example, the inferences and observations in sec 4.1 and sec 5 are conditioned on the assumption that the factors of variation are known, which gives us a cue to which model might work (CycleGAN for low data-drift, pretraining for unseen data shift etc). But can this knowledge be extrapolated to judge *in-the-wild* datasets? This brings me to the next point.

- I could not pinpoint a single take-home message from the paper. The empirical study results have high variance, and it can only be inferred that no single method works for all the cases. But can the authors, equipped with the knowledge of these experiments, draw any *meta recommendation* of mapping between (factors of variation, distribution shift, modeling) choices? Of course, I fully agree that the current findings are still incredibly useful, but I was expecting to see a more optimistic take home message for the readers : ) .
- I fail to see that significance of the latent factorization model proposed in sec 2.1. While the formulation of the data, discrete attributes and the labels seem clear, I do not see the significance of introducing the latent factor *z*. The explanation seems equally effective by using only notation of (x,y). Also looking at the models used to achieve robustness, none of weighted resampling or data augmentation require any latent factorization. Similarly, Imagenet pretraining also does not necessarily relate to latent factors in the attributes. Beta-VAE however relates to latent factors but that's about it. Perhaps the whole section 2.1 can be better motivated in the context of the paper.
- The domain adaptation works generally work with a *covariate shift* assumption [2]. How does that fit into the proposed framework? Also, DA works like DANN and CORAL generally make use of additional unlabeled data. How are the unlabeled data chosen for the experiments? While it is inferred (sec 4.2) that domain adaptation methods lead to limited improvements, the DA methods considered are quite old and primitive. Perhaps more recent DA methods might help?
- The authors could also contrast the in-distribution vs. out-of-distribution performance of the models. It seems that arch. like ViT might do very well on training domain but lack generalization on OOD test data. Why is this so? Does this have to do with the strong prior that CNNs induce that help generalization?
- Is Imagenet pretraining performing better only because it is trained on larger scale data compared to other methods? In Fig 7, are all models trained/pretrained using same amount of data?


1. Hendrycks, Dan, and Thomas Dietterich. "Benchmarking neural network robustness to common corruptions and perturbations." _(2019).
2. Ben-David, Shai, et al. "A theory of learning from different domains." _Machine learning_ 79.1 (2010): 151-175.

**Summary Of The Paper:**

## Very useful study on calibrating OOD models and benchmarks

This paper provides a fine-grained analysis on nature of distribution shifts in image datasets through extensive experiments on various benchmark methods for robustness. They work with an assumption that the underlying factors of variation, including the label, are encoded through discrete attributes. The authors provide a concrete framework to put forth the latent factorization model, including appropriate grouping for various types of benchmark methods, distribution shifts as well as training conditions. Several experimental demonstrations are provided to compare and contrast robustness of existing methods leading to numerous useful observations.

**Summary Of The Review:**

Overall, I believe that the experiments conducted as part of this paper are very well structured and provide multiple useful insights to the community. However, the analysis still falls short in several places (see above). The latent factorization models needs to be better motivated, and the generality of the framework needs to be clarified. Still, this is a very useful work to the community, and if the authors could clarify the questions raised, I would be happy to update the score. I haven't gone through the supplementary material in detail. If any of the questions above have a direct answer in the suppl. material, the authors can directly point to that and I would be happy to update my comments.

---

> ### Author Response · Authors · 2021-11-16
> **Rebuttal for Question 7 for Reviewer HwsU**
>
>
> - *Question 7: Is Imagenet pretraining performing better only because it is trained on larger scale data compared to other methods?*
>
> It is presumably because ImageNet is trained with more data. However, we note that this does not always improve performance much (as in Figure 10 for iWildCam and Camelyon) and for small values of N in Figure 11 for iWildCam and Camelyon). Therefore it seems to depend on how much the features learned on the pretrained data are helpful for the downstream task. In Fig7, all but pretraining on ImageNet are trained with the same amount of data. We clarify this in section 4, paragraph additional data.
>
> [1] Goel et al. “Model Patching: Closing the Subgroup Performance Gap with Data Augmentation”
>
> [2] Gulrajani et al. “In search of lost domain generalization.”
>
> [3] Sagawa et al. “Distributionally robust neural networks for group shifts: On the importance of regularization for worst-case generalization.”
>
> [4] Li et al. “Domain generalization with adversarial feature learning”

---

> ### Author Response · Authors · 2021-11-16
> **Rebuttal for Questions 3-6 for Reviewer HwsU**
>
> - *Question 3: I fail to see that significance of the latent factorization model proposed in sec 2.1.*
>
> We agree with the reviewer that in principle we can integrate over the latents and describe the marginal model $p(X, Y)$ without a reference to a latent $z$. However, from a modelling point of view, we identify several benefits for including $z$.
>
> First, we use this formulation to characterize different approaches and why they should encourage generalization. There are two classes of models that rely on this latent factorization in addition to the B-VAEs. This is useful to understand why learned data augmentation can be helpful (e.g. in CAMEL [1], section 3, paragraph **learned data augmentation**) and how access to the true generative model would allow us to solve these distribution shift problems. Additionally, we can use it to characterise domain generalization methods and how they are trained for invariance to the irrelevant attribute (again in section 3, paragraph **domain generalization**).
>
> Another benefit of introducing a latent representation is that , we can use it to model the case that $y$ is potentially a non-exhaustive list of the factors of variation. As a result, x is not deterministic given the set of attributes. We use a hidden representation $z$, conditioned on which, the generating process of images $p(x|z)$ is invariant across different domains. This additionally allows us to describe the true underlying generative process and differentiate it from the one conditioned on the given distribution of attributes: $p(x|y^{1…k})$. While $p(x|y^{1…k}), p_\textrm{train}(x|y^{1…k}),  p_\textrm{test}(x|y^{1…k})$ may vary, but $p(x|z), p_\textrm{train}(x|z)$ and $p_\textrm{test}(x|z)$ do not.
>
> - *Question 4: Also, DA works like DANN and CORAL generally make use of additional unlabeled data.*
>
> We apply domain adaptation approaches by treating different attribute values of the nuisance variable as different domains and then enforcing that the features learned between these domains are invariant, similar to [2]. We could operate precisely as those methods were originally described and throw away labels but, in our case, this would unfairly penalise DANN and CORAL. (We do not compare methods in the precise setting where we have data from two domains at train time: one labelled and one unlabelled. We also do not evaluate test time adaptation which is an interesting area, but we left beyond the scope of the current paper.) We clarify this in section 3 (paragraph **domain generalization**).
>
> Our results seem to corroborate findings elsewhere (e.g. [2]) that find that most domain generalization methods do not help over ERM; however, we explore this on multiple distribution shifts and amounts of shift. We are happy to include more methods in our framework. We have added GroupDRO [3] and the MMD loss [4] into our framework and are currently running these methods on a subset of our datasets and shifts (unseen data shift on dSprites and Camelyon). We will update with results as they come in.
>
> - *Question 5: The domain adaptation works generally work with a covariate shift assumption.*
>
> The covariate shift assumption is valid as the distribution over labels given a sample within any attribute value should be the same.
>
> - *Question 6: The authors could also contrast the in-distribution vs. out-of-distribution performance of the models.*
>
> This is an interesting question. We find that there can be large generalization gaps for ViT and MLPs but this is not consistent across the two real-world datasets:
> - For Camelyon, we actually do find that ViT and MLP both have higher in-distribution validation scores than the ResNet models (>90% for MLP/VIT vs ~65% for the ResNets). However, when we look at the out-of-distribution test set, their performance is much worse (57% for MLP/ViT vs 64% for the ResNets). This problem is not solved by model selection (see Figure 12, which compares using the in-distribution and out-of-distribution validation set for model selection on iWildCam and Camelyon).
> - However, on iWildCam, the story changes. Here the in-distribution validation scores for ViT are around 66% whereas for the ResNet they’re around 63%. This translates to the out-of-distribution test accuracy, where the ViT model performs best.
>
> It’s not clear why precisely ViT works better on iWildCam and doesn’t overfit. We hypothesise that the reason ViT may do better is due to its specific properties. The backgrounds are static across different locations, so both models must learn to ignore these values in order to estimate anything of use. For Camelyon, the ViT model may learn features that are too specific, whereas the CNN uses inductive biases on spatial invariance to generalise better. What precisely causes these models to generalise differently and how they operate in different settings is an interesting question that our framework can be used to address in the future.

---

> ### Author Response · Authors · 2021-11-16
> **Rebuttal for Questions 1-2 for Reviewer HwsU**
>
> We thank the reviewer for their thorough and insightful review. We are glad they appreciate our work as a useful contribution to the community. We have made the changes suggested and have answered questions below. We have split the answers into multiple comments.
>
> - *Question 1: My major question to the authors of the paper is how general the proposed framework?*
>
> Our framework was evaluated on large in-the-wild datasets, allows for the consideration of multiple attributes including unobserved, entangled ones, and it can be used to explore more complex distribution shifts than those based on two observable attributes. We discuss this further below.
>
> Camelyon and iWildCam **are** large in-the-wild datasets (with hundreds of thousands of images, so of a similar scale to ImageNet) and explore real-world problems (e.g. tumour detection across different hospitals and detecting animals in camera trap imagery across different countries). So, conclusions could most likely be extrapolated to similar problems or setups (e.g. other medical imaging tasks for Camelyon). Overall, we appreciate the hesitancy in extrapolating to markedly different settings (e.g. ImageNet) where the challenges and properties may be different but we believe that similar problems and setups to those investigated would exhibit similar results (e.g. similar medical imaging tasks for Camelyon).
>
> Also, our framework is more general than just considering two attributes. All the datasets were annotated with additional labels and attributes but there is **no** assumption that all factors of variation are known. Indeed in Camelyon and iWildCam this is the case and there is no assumption that the attributes labelled are not entangled with other unobservable factors of variation (as indeed may be the case in Camelyon and iWildCam).
>
> Finally, our framework for creating shifts allows for more complex distribution shifts than those used in the paper. We can define distribution shifts using a comparison operator on one or more labels to create a distribution shift. Shifts can then be composed together to create more complex shifts between train and test (see Appendix E.1-E.2). This allows the exploration and evaluation of multiple entangled factors of variation.  Finally, there is no necessity for methods to make use of the additional attribute information beyond the label, which allows the evaluation of unseen factors of variation (by not allowing methods to use knowledge of a given factor).
>
> - *Question 2: I could not pinpoint a single take-home message from the paper.*
>
> We thank the reviewer for their comments and agree that it would be great to know, for a given dataset, what precise method to choose a-priori. However, we find that the conclusions can vary based on the particular attribute being considered (Appendix B.3) and dataset (Figures 10-12). Moreover, the choices may depend on the precise dataset properties (for example, are transformations arising from image-to-image transformations versus other types of change). Models may make implicit assumptions that are subtle to disentangle. Therefore, we hesitate to give one single meta-recommendation that may not generalise to an unseen dataset or attribute. However, our framework can be used to run similar experiments or to infer how models will perform on similar datasets and distribution shifts. We do raise this question in section 5 in order to encourage other researchers to investigate these problems and define concrete actions.

---

> > ### Comment · Reviewer_HwsU · 2021-11-18
> > **Response to author response 1**
> >
> > Thanks for your replies.
> >
> > Question 1: I see. Thanks for clarifying that the assumption that only two factors of variation are present is not as restrictive as I assumed. Perhaps you can add little detail on this aspect into the main paper.
> >
> > I now also understand the reasoning behind the latent factorization modeling. Strictly speaking it can be skipped, but if it is helping the modeling aspect, then it makes sense to explain it as such.
> >
> > Question 4: The experiment setting is still not clear to me. DA methods require, by definition, a source domain with labels and target domain with unlabeled data. So can you explain this statement "We could operate precisely as those methods were originally described and throw away labels but, in our case, this would unfairly penalise DANN and CORAL. (We do not compare methods in the precise setting where we have data from two domains at train time: one labelled and one unlabelled.)"

---

> > > ### Author Response · Authors · 2021-11-19
> > > **Response to reviewer HwsU's 2nd response**
> > >
> > > # Answers to questions
> > >
> > > *Question 1: I see. Thanks for clarifying that the assumption that only two factors of variation are present is not as restrictive as I assumed. Perhaps you can add little detail on this aspect into the main paper.*
> > >
> > > We have added a reference at the start of Section 2.2 to Appendix E with the sentence “Our experimental framework can be used to compare more complex shifts, discussed in Appendix E.”
> > >
> > > We have also added a couple of additional sentences in Appendix E discussing this: “Our framework for creating shifts allows for creating complex distribution shifts. We can define distribution shifts using a comparison operator on one or more labels to create a distribution shift. Shifts can then be composed together to create more complex shifts between train and test. This allows the exploration and evaluation of multiple entangled factors of variation. Finally, there is no necessity for methods to make use of the additional attribute information beyond the label, which allows the evaluation of unseen factors of variation (by not allowing methods to use knowledge of a given factor).”
> > >
> > > *Question 4: The experiment setting is still not clear to me. DA methods require, by definition, a source domain with labels and target domain with unlabeled data. So can you explain this statement "We could operate precisely as those methods were originally described and throw away labels but, in our case, this would unfairly penalise DANN and CORAL. (We do not compare methods in the precise setting where we have data from two domains at train time: one labelled and one unlabelled.)"*
> > >
> > > We follow [1] who apply these methods to a domain generalization setup. Methods such as DANN typically have two losses: (1) one is on the labelled domain to minimize prediction error and (2) the second is to bring the two domains closer together by enforcing some form of invariance in feature space (for example DANN uses an adversary to learn a representation such that the adversary cannot distinguish between the domains). [1] apply this in the domain generalization case (where they have N domains and labels for all domains but want to generalise, *at test time*, to an unseen domain) by applying loss (1) to all domains and loss (2) across domains to promote invariance of the features to the different domains. This should, hopefully, improve generalization to new domains.
> > >
> > > In our case, we treat the different values of the nuisance attributes as different domains and proceed as follows. For all values of an attribute, we apply loss (1). We apply loss (2) across the different values of the attribute to enforce that the learned representation $z$ is invariant to the value of the nuisance attribute. If $z$ were invariant, the learned representation should be able to generalise in all shifts considered. We note that with the code, this implementation of the methods will be obvious. Please let us know if this is still unclear.
> > >
> > > [1] ​​Gulrajani et al. “In search of lost domain generalization.”
> > >
> > >
> > > # Additional Results
> > >
> > > We have run MMD and GroupDRO on dSprites, MPI3D, and Camelyon on the unseen data shift and low data shift settings. Neither method performs significantly better on these datasets than the methods explored in the paper. The only exception is MMD on Camelyon, which achieves 79% (+/-2) top 1 accuracy on Camelyon (+8% better than the next best method: 71% (+/- 1)). This result is obtained using the in distribution validation set. However, if we look at performance on the out of distribution validation set (which was never seen during training), results change. The top 1 accuracy for MMD is 72% (+/-3) versus DANN which obtains 74% (+/- 1).
> > >
> > >
> > > These results corroborate our conclusions that: (1) we can do better than the ERM method (**takeaway 1**); (2) results vary due to the domain (**takeaway 6 and 7**); and (3) that while domain generalization algorithms can improve performance, it is not clear a-priori which ones will, on which datasets (**discussion point 1**) and how to make these algorithms useful generically (**tip 4**) without trying all options.

---

> > > > ### Comment · Reviewer_HwsU · 2021-11-20
> > > > **Response to author's response to my response on the authors response**
> > > >
> > > > Cool! I think all my questions were answered. Although I think using unsupervised domain adaptation algorithms for domain generalization is odd, when dedicated algorithms do exist towards domain generalization [1,2,3]. And keeping in mind that this is purely a empirical paper with reasonable technical novelty and the fact that the eventual observations and inferences are partly expected and partly surprising (from sec 4.1 and 4.2 ), I settle with a score of 8.
> > > >
> > > > 1. Dubey, Abhimanyu, et al. "Adaptive Methods for Real-World Domain Generalization." Proceedings of the IEEE/CVF Conference on Computer Vision and Pattern Recognition. 2021.
> > > >  2. Li, Ya, et al. "Deep domain generalization via conditional invariant adversarial networks." Proceedings of the European Conference on Computer Vision (ECCV). 2018.
> > > >  3. Li, Da, et al. "Deeper, broader and artier domain generalization." Proceedings of the IEEE international conference on computer vision. 2017.

---

### Decision · Program_Chairs · 2022-01-20

**Decision:**

Accept (Oral)

**Comment:**

The paper proposes a general framework to reason about fine-grained distribution shifts, evaluating a large set of different approaches in a variety of settings. All reviewers recommend acceptance. While concerns were raised, including questions about the generality of the framework, unsurprising “tips”, and unclear take-home messages, all reviewers find the work strong, with an elegant formulation, and useful insights. The AC agrees with the reviewers that this work addresses a very important problem, proposes an interesting unified framework and benchmark for domain shift analysis, and should be a valuable tool for the community to pursue further research in this area.